# A Simulation-Data-Based Machine Learning Model for Predicting Basic Parameter Settings of the Plasticizing Process in Injection Molding

**DOI:** 10.3390/polym13162652

**Published:** 2021-08-10

**Authors:** Matthias Schmid, Dominik Altmann, Georg Steinbichler

**Affiliations:** 1Institute of Polymer Injection Moulding and Process Automation, Johannes Kepler University Linz, Altenberger Straße 69, 4040 Linz, Austria; dominik.altmann@jku.at (D.A.); georg.steinbichler@jku.at (G.S.); 2Kompetenzzentrum Holz GmbH (Wood K Plus)—Biobased Composites and Processes, Altenberger Strasse 69, 4040 Linz, Austria

**Keywords:** machine learning, multilayer perceptron, neural network, regression, plasticizing, polymers, basic settings, prediction, simulation, data-based, model, quality

## Abstract

The optimal machine settings in polymer processing are usually the result of time-consuming and expensive trials. We present a workflow that allows the basic machine settings for the plasticizing process in injection molding to be determined with the help of a simulation-driven machine learning model. Given the material, screw geometry, shot weight, and desired plasticizing time, the model is able to predict the back pressure and screw rotational speed required to achieve good melt quality. We show how data sets can be pre-processed in order to obtain a generalized model that performs well. Various supervised machine learning algorithms were compared, and the best approach was evaluated in experiments on a real machine using the predicted basic machine settings and three different materials. The neural network model that we trained generalized well with an overall absolute mean error of 0.27% and a standard deviation of 0.37% on unseen data (the test set). The experiments showed that the mean absolute errors between the real and desired plasticizing times were sufficiently small, and all predicted operating points achieved good melt quality. Our approach can provide the operators of injection molding machines with predictions of suitable initial operating points and, thus, reduce costs in the planning phase. Further, this approach gives insights into the factors that influence melt quality and can, therefore, increase our understanding of complex plasticizing processes.

## 1. Introduction

Finding optimal parameter settings for the plasticizing process (see Section 2.1—“The Plasticizing Process and S3 Simulation Software”) is one of the most important tasks in operating polymer processing machines. In injection molding and extrusion, the goal is to determine an operating point that satisfies all melt quality and machine lifetime requirements. The most relevant parameters with the highest impact on melting behavior are the pressure, screw rotational speed, and cylinder temperatures [1].

Especially in injection molding, much information is available about the early product cycle stages of the process. In this paper, we wanted to push the approach of a simulation-driven data-based model as we found that simulations have become increasingly used for the screw layout and process optimization. This valuable information could also be employed to determine basic machine settings. From personal experience and collaborating work, we observed that many operators adjust the plasticizing parameters for process stability but without additional knowledge about the current process. Due to the complex melting behavior—depending on the molecular weight, molecular weight distribution, chain branching, shear rate, and shear stress—of polymers, we found that it is not known exactly whether a selected operating point is efficient [2].

A data-based digital twin (virtual representation) of the plasticizing process (physical object) that “knows” the correlations between the melt quality and plasticizing parameters (predicting the performance of a physical twin) could, therefore, be beneficial [3]. A simulation-data-based model could already be built in the screw-selection phase. Given the boundary conditions of the main process, such as the material, screw geometry, and maximum cycle time, a digital twin could assist the operator by predicting relevant basic parameter settings that require little optimization.

Research has focused intensively on machine-learning models of the injection process to predict quality parameters, such as the weight or dimensions of parts [4,5,6]. However, one of many problems that influence the final part quality can already occur one step earlier, that is, in the plasticizing process, often due to insufficient melt quality.

This paper describes the development of a supervised regression model that—given minimal input information—is able to predict the basic settings for the plasticizing process. The generation and preprocessing of a simulation-based data set are explained in detail. We further describe the process of building an artificial neural network (multilayer perceptron), discuss its accuracy, and compare the results of experiments with those using the predicted basic settings.

## 2. Basics

This paper describes the workflow to construct a regression model that can predict the basic machine setup for the process parameters of the back pressure and screw rotational speed. Additionally, the approach for the experimental evaluation will be explained in detail. All distributions and parameter values shown in this section are based on the material “PP-HE125MO” and a three-zone screw with LD 20 and a 30 mm diameter.

### 2.1. The Plasticizing Process and S3 Simulation Software

As already mentioned, the plasticizing unit is one of the most important functional components of an injection molding machine with the task to feed in solid material and melt it along the screw length. The unit usually includes a barrel combined with a specific reciprocating single screw, a drive, and heating bands. A typical setup is shown in Figure 1.

Three main functional zones can be identified that are responsible for solid conveying, melting, and melt conveying. To gain better insights into these processes, a software tool called S3 (screw simulation software) [8] was developed with the aims of (i) predicting the screw geometries and process parameters that are optimal in terms of part quality and the machine lifetime for a given application and (ii) finding a good compromise between the computational power required and model accuracy. The simulation times achieved with this software can be as short as one minute.

The input parameters include the barrel and screw geometries, materials, and process and simulation parameters. The output parameters include the plasticizing rate and time, power consumption, pressure build-up, temperature distribution, and melting behavior. The latter parameter is defined in the simulations by a melting curve (percentage of molten material) along the length of the screw and measured visually during the experiments.

Commercial software solutions are often based on analytical models or are not developed for the discontinuous plasticizing process as in injection molding. The main advantage of the S3 software is the flexibility regarding its easy implementation of new material and calculation models or various new approaches. A detailed description of the S3 software and its comparison with other software can be found in [8].

### 2.2. Artificial Neural Networks

Artificial neural networks are currently one of the most important supervised machine learning methods, characterized by the presence of labels (i.e., target values) that can be either numerical data (task: regression) or categorical data (task: classification) [9].

The simplest form of a neural network consists of only two layers—an input layer and an output layer. This specific case is called a linear perceptron, since it can distinguish only linearly separable data. However, in real life, many problems can only be modeled non-linearly, and neural networks with at least one hidden layer, and non-linear activation functions (called multilayer perceptrons) were, therefore, introduced [10].

More complex models with many hidden layers are called deep neural networks (there is no consensus on the number of hidden layers required to use this term). The network built in this work consists of three hidden layers and is defined as a multilayer perceptron. There are two phases in training a neural network: the forward pass and the backward pass. Figure 2 shows a schematic representation of the forward pass of a multilayer perceptron with one hidden layer. The sizes (i.e., the numbers of neurons) of the input and output layers are defined by the data to be processed.

In the first phase, the inputs are moved forward in the output layer direction. Every neuron in the hidden layer has one pre-activation $s_{i}$ and one activation $a_{i}$. The forward pass, which can be interpreted as the prediction of the output, is illustrated in more detail for the neuron in the red box in Figure 2. The pre-activations are calculated by:(1)$$s_{i}=\sum_{j=1}^{Q}w_{ji}x_{j}+b_{i}.$$

This gives the linear sum of three products of the input neurons $x_{j}$ with their corresponding weights $w_{ij}$ plus a bias term $b_{i}$. Depending on the non-linear activation function used, the pre-activation $s_{i}$ is, then, mapped to $a_{i}$:(2)$$a_{i}=f\left(s_{i}\right).$$

In the next step, the activation $a_{i}$ of the neuron serves as input to the next layer, and the procedure is repeated until the end of the output layer of the network is reached [11].

The network is trained in the backward pass, where all weights and biases are updated. This is normally done via gradient-descent methods after computing the error (i.e., loss) in the output layer between the prediction (forward pass) and real label. The error is backpropagated from the last layer through the hidden layer and finally to the input layer [12].

## 3. Methods

We describe the workflow for constructing a regression model that can predict basic machine settings—that is, the back pressure and screw rotational speed—for an injection molding plasticizing process. All distributions and parameter values shown in this section are based on the material “PP-HE125MO” and a three-zone screw with an LD ratio of 20 and a diameter of 30 mm.

### 3.1. Data Generation and Preprocessing

Figure 3 presents a flowchart of the neural network construction. The first branch at the top left illustrates the input parameters for the simulation and the outputs that are generated. The groundwork for the multilayer perceptron model was laid by simulating 2000 data points with the S3 Software.

The design points were chosen by keeping all simulation-relevant input parameters (e.g., grid points and time steps) constant while varying the process parameters shown in Table 1 between the minimum and maximum values. The short S3 simulation times made finding a more efficient way of building the data set (e.g., using the design of experiment method) unnecessary. The amount of simulation data points can be decreased significantly with adequate domain knowledge, since non-feasible input values would either not converge in the simulation or would be filtered by preprocessing of the model, which is described below.

Further important process parameters include the feed-trough and cylinder temperatures, which were—for simplification—considered to be constant at 60 and 240 °C, respectively.

The main challenge in the prediction of the back pressure and screw rotational speed is that these parameters are used as an input to the simulations. It is important to understand that the outputs of the simulation cannot be used directly to predict these parameters for basic machine settings. Hence, a model that only reproduces the simulation is unhelpful. Therefore, at this point, the following questions must be answered:Which features (inputs) can be selected from the data in order to predict the desired labels?How can the model fulfill the requirement of good melt quality for the predictions?

A crucial feature is the shot weight, which can be derived from the mass flow rate and the plasticizing time. In our approach, the melt quality is measured by the percentage of molten material along the screw length. For example, the screw position (in length-to-diameter ratio; LD; melt quality—feature 1) at which 99% of the material is molten (melt quality—feature 2) can be determined and used as input to the model in the form of two features. The fourth feature for predicting the back pressure and screw rotational speed is the corresponding plasticizing time, which is directly given by the simulation output. Hence, the input of the model is defined by the following features:shot weight;melt quality—LD screw position;melt quality—molten material [%]; andplasticizing time.

We found that the simulation input parameters cycle time and screw starting point had a negligible impact on the model performance and, therefore, need not be considered. Information about the metering stroke is included in the shot weight. The distributions of all parameters of the raw data set are shown in Figure 4. Since the simulation input values were drawn randomly (see Table 1), the distributions are well balanced within their limits. However, the distributions of the simulation outputs—melt quality (melt value and LD) and plasticizing time—are highly unbalanced.

The simulation output information about melt quality is given by a large array that describes the melt percentage along the screw length. The important samples in our data set are those with good melt quality. For each sample, the LD screw position at which 99% of the material is molten was, therefore, determined and extracted into the data set. However, numerous data points remained that did not fulfill this requirement. Apparently, the screw positions of all these samples are at the very end of the screw (LD 20.5), which can be seen in the top right distribution in Figure 4.

The requirement of 99% molten material makes the distribution curve of the melt value relatively unbalanced. All samples with a melt value greater than 0.01 (<99%) correspond to the screw position LD 20.5.

To ensure that the model is trained only by samples that guarantee good melt quality, problematic data points were discarded. Given the distributions of the raw data set, this was easily achieved by discarding all samples with a screw position equal to 20.5 LD.

Due to the requirement to predict only operating points with good melt quality, the model was not trained with “bad” samples. This filtering process reduced the data from 2000 to 915 samples. The corresponding distributions (Figure 5) show that the data set was much more balanced, which was also beneficial for training the multilayer perceptron.

### 3.2. Model Construction

Numerous supervised machine learning methods are available for building a model that can handle data sets with complex non-linearities. We trained models using prevalent machine learning algorithms. We compared the following methods:multilayer perceptron,Gaussian process regression,support vector regression,polynomial regression,random forest, andgradient boosting.

Since the multilayer perceptron outperformed all other methods (see Section 4—“Results”), we explain the model construction for this method only. The neural network was implemented with Python’s [13] open-source library Pytorch [14], using the following architecture and hyperparameters:Training set: 549 samplesValidation set: 183 samplesTest set: 183 samplesLayer structure: 4 –> 50 –> 50 –> 30 –> 2Activation function: Tanh (for all layers)Optimizer: Pytorch AdamLoss function: Pytorch MSELearning rate epoch 0–600: $10^{-3}$Learning rate epoch 600–1200: $10^{-4}$Learning rate epoch 1200–1500: $10^{-5}$Weight decay: $10^{-4}$Batch size: 32Epochs: 1500.

This specific setting was found with help of a hyperparameter study. Fewer epochs would also have resulted in a good model; however, the small data set (compared to image data sets) allowed fast training. Overfitting was only detected with a much larger number of trainable parameters.

### 3.3. Experimental Evaluation of the Model

Validating the model performance with results from a real injection molding machine required experiments to be developed. Two basic parameter settings (the screw rotational speed and back pressure) were predicted by the neural network model with the following ranges of feature values:melt value: 99% molten for each data point;screw Position (LD): 16, 18, 20;shot weight (kg): 0.02, 0.035, 0.05; andplasticizing time (s): 1–15.

The workflow is described in Figure 6. Since the multilayer perceptron model cannot outperform the simulation it is built on, its predictions must be validated with data points produced by a real machine.

For every combination of LD and shot weight, 40 samples with increasing plasticizing times were created, which resulted in a data frame of 3 × 3 × 40 = 360 samples. All 360 samples served as input to the model, which predicted the corresponding parameter settings in the forward feed. Since the simulations were limited to the ranges 25–225 bar for the back pressure and 0.2–1 m/s for the screw rotational speed, the predictions of the 360 samples had to be filtered to discard all non-feasible data points.

Table 2 lists the parameters of the experiments, where the shot weight and melt value were 0.035 kg and 99% for all samples.

## 4. Results

To identify the most suitable modeling approach for our purpose, we compared various supervised learning methods in terms of their performance. Table 3 shows the overall absolute mean errors in percentages and the corresponding standard deviations for the two labels back pressure and screw rotational speed for both the training and the test sets. The algorithms are listed in order of decreasing performance and for the sake of completeness, all hyperparameters of the corresponding best model are provided in the Appendix A. A low mean error on the training set and a much higher error on the test set indicates overfitting.

This means that the model can reproduce already seen data (i.e., training data) very well, while its prediction of unseen data (i.e., test data) is poorer. This was especially the case for Gaussian process regression and for polynomial regression. Decision-tree methods—random forest and gradient boosting—were unsuitable for this data set when the settings from the hyperparameter search were used. Overall, the multilayer perceptron outperformed all other methods on the given data set, as it exhibited a markedly lower generalization error on the test set.

### 4.1. Results—Multilayer Perceptron Model

With increasing complexity, neural networks tend to overfit to training data. The hyperparameters (e.g., learning rate) must, therefore, be tuned such that the generalization risk error (i.e., the error on the test set) is kept low. Figure 7 plots the losses of the training and validation sets. Both losses decreased steadily until reaching a low plateau, which indicates a generalized model. The loss was calculated in a loop over all epochs for the corresponding data sets and was aggregated over the batch sizes.

Therefore, with the same batch size, but varying lengths of the training and validation set, the resulting loss for the validation set could be lower than for the training set. At epoch 600, the learning rate decreased from $10^{-3}$ to $10^{-4}$ and, at epoch 1200, to $10^{-5}$. Decreasing the learning rate is a commonly used approach because it allows greater weight changes in the beginning of the training phase and smaller changes at the end [15].

Table 4 lists the model performances for the training and test sets for the two labels back pressure and screw rotational speed. Regression models are usually evaluated by the mean squared error. For better interpretation, we chose the root-mean-squared error as a metric. As expected from the loss curves, the errors of the labels were very low for both data sets. This indicates good generalization and shows that the model predicted all simulation data points almost perfectly within the chosen limits.

Figure 8 and Figure 9 visualize the values of the input parameters back pressure and screw rotational speed for all data points. During the training phase, the model learned only from the blue samples, and, for the hyperparameter search, the green unseen validation data points were taken. During the evaluation phase, the generalization error was determined with the unseen data points of the test set. As explained in the Methods section, all samples achieved good melt quality at screw positions between LD 16 and LD 20.

The multilayer perceptron model predictions, illustrated by a black cross in Figure 8 and Figure 9, provide further evidence of the good generalization of the model to unseen data (validation and test sets). Note that the training set predictions were very accurate, while the validation and test set predictions were slightly poorer for some specific samples. However, the deviations of the predictions of unseen samples were sufficiently small to ensure a well-generalized model for both labels.

### 4.2. Results—Model vs. Experiment

We established that the simulation can be accurately described by the neural network model. However, our main objective was the prediction of settings for the back pressure and screw rotational speed given the boundary conditions of a specified melt quality and plasticizing time at a selected screw position.

Figure 10 plots the errors in plasticizing time—given as the mean and standard deviation for each sample—for three experimental runs performed respectively with the materials PP-HE125MO, PEHD-MB7541, and PA6-B30S. The materials were fully characterized at our institute in regard to all relevant rheologic and thermodynamic material parameters that were required for the simulations. We, therefore, assume that differences between the model and experiment were not caused by inadequate material models. The plasticizing error is illustrated by the mean and standard deviation of three measurements for each sample.

The experimental results (see Table 5) show that the predictions of the basic parameter settings were good for the PEHD-MB751 material, with an average absolute error of 2.8%, an absolute standard deviation of 2%, and a maximum error of 8%. For this material, our approach produced suitable machine settings. For PA6-B30S, the absolute mean error was 10.8% with a standard deviation of 6% and a maximum error of 18%. For PP-HE125MO, the prediction performance was poorer, with an absolute mean error of 14.5%, a standard deviation of 10%, and a maximum error of 34%.

Note that the errors, illustrated in Figure 10, are due mainly to the simulation not yet being able to consider machine behavior, such as material feeding and conveying of solid material. It appears that machine behavior plays a decisive role in the prediction of PP-HE125MO, since we observed considerably greater errors between the simulated and real torques.

### 4.3. Conclusions and Outlook

We presented a workflow for constructing a simulation-data-based multilayer perceptron model that is able to predict settings for the plasticizing parameters back pressure and screw rotational speed to result in operating points with good melt quality (fully melted material). We demonstrated that, after feature extraction and further preprocessing of the data set, the input variables—screw position where 99% of the material is molten, plasticizing time, and shot weight—were sufficient to provide a reliable, generalized model. The filtered data set comprising 915 simulation data points was split into training, validation, and test sets. The overall performance of the simulation model (digital twin) was assessed by calculating the root-mean-squared error and was visualized in plots. The small error on the test set indicates a low generalization error and, therefore, good performance on unseen data.

For further evaluation of our approach, we conducted experiments with three different materials at the predicted operating points and determined the difference between the real and desired plasticizing times. The melt quality was estimated visually and was acceptable in all cases. The average absolute errors between the real and desired plasticizing times were 2.8%, 10.8%, and 14.5% for PEHD-MB7541, PA6-B30S, and PP-HE125MO, respectively. These errors can be attributed to differences between simulation and reality that arise mainly from machine behavior and the material used. For PEHD, the prediction agreed well with the experimental result; however, for PP, the errors were larger because of machine behavior (increased back pressure). The overall accuracy, however, was high enough to obtain a suitable starting point for optimizing the machine settings.

In the future, given the continuous improvements in simulation accuracy, data-based machine learning models will provide even better assistance to operators in choosing suitable basic machine settings. The errors caused by machine behavior could be minimized by building a second model that includes experimental samples or by updating the existing model by means of transfer learning methods [4,6]. Incorporating cylinder temperatures into the predictions will require more complex models, which is another possible avenue for future research.

## Figures and Tables

**Figure 1 polymers-13-02652-f001:**
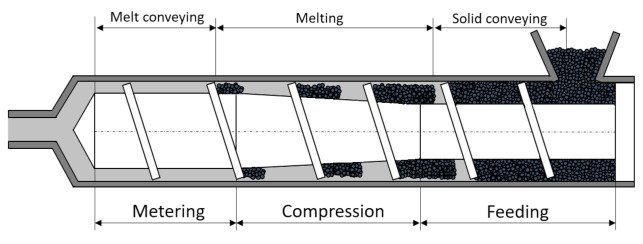
Schematic representation of the functional zones of a plasticizing unit [7].

**Figure 2 polymers-13-02652-f002:**
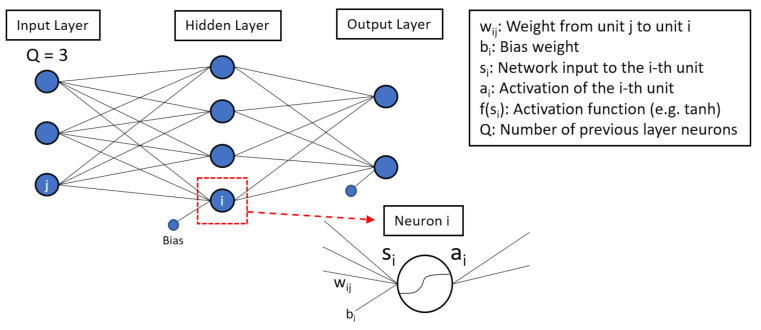
Forward pass of a multilayer perceptron. The red box shows the determination of the pre-activation and activation in one neuron of the hidden layer. The pre-activation is calculated by the linear sum of the product of all previous neurons $x_{j}$ (input layer) with their corresponding weights $w_{ij}$ plus a bias term $b_{i}$. The pre-activation $s_{i}$, then, serves as input to the non-linear activation function, which gives $a_{i}$.

**Figure 3 polymers-13-02652-f003:**
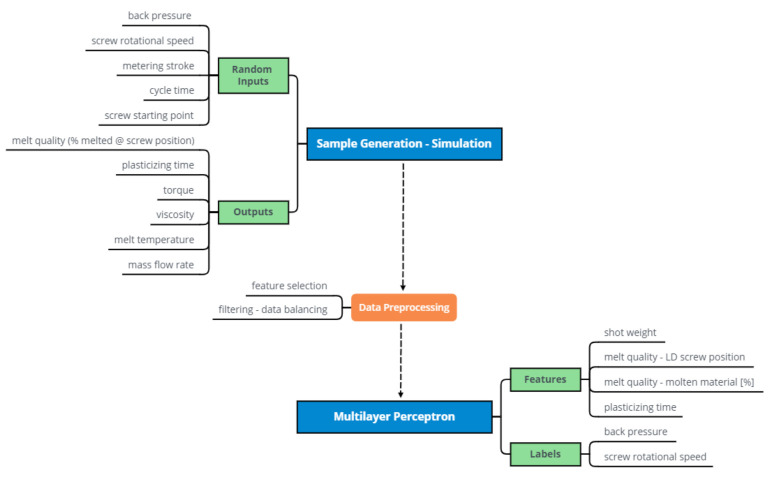
Flowchart—The construction of a neural network model (multilayer perceptron).

**Figure 4 polymers-13-02652-f004:**
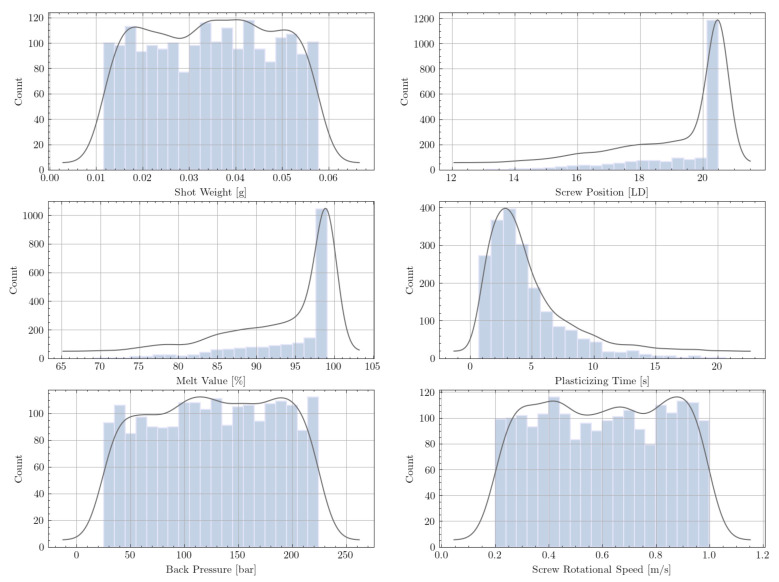
Distribution of the features and labels for the raw data set (2000 samples).

**Figure 5 polymers-13-02652-f005:**
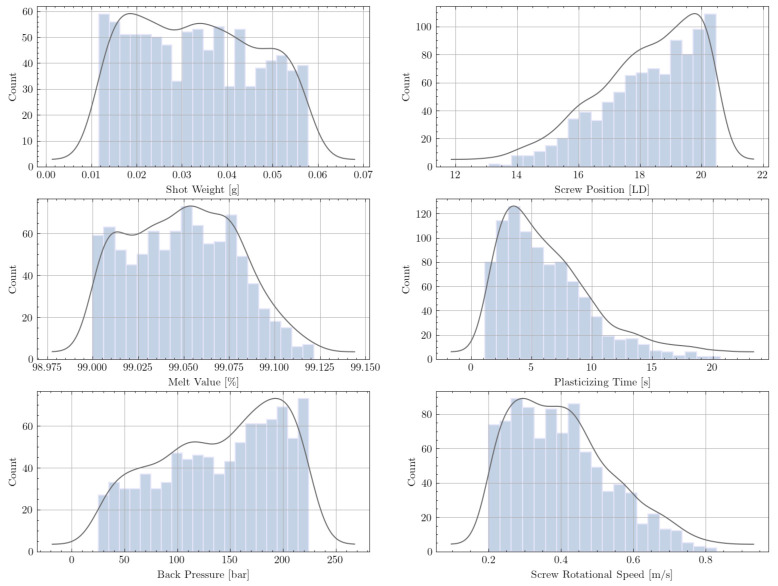
Distribution of the features and labels for the filtered data set (915 samples).

**Figure 6 polymers-13-02652-f006:**
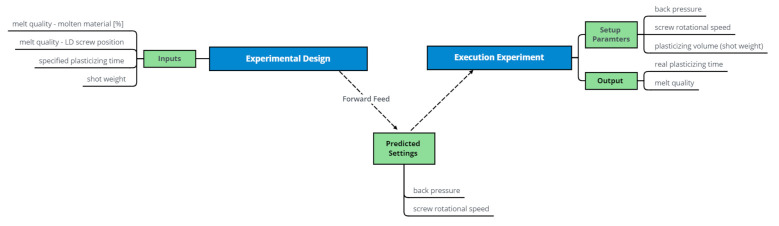
Flowchart—Experimental evaluation of the model.

**Figure 7 polymers-13-02652-f007:**
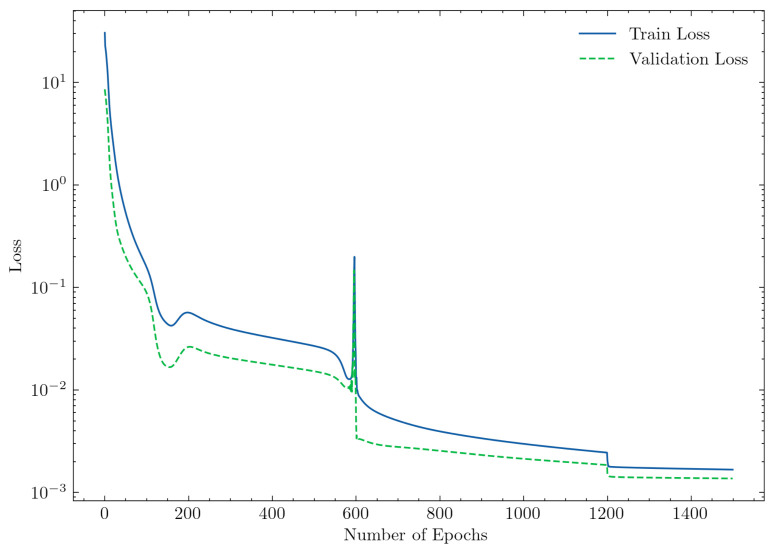
Losses on the training and validation data sets in the training phase.

**Figure 8 polymers-13-02652-f008:**
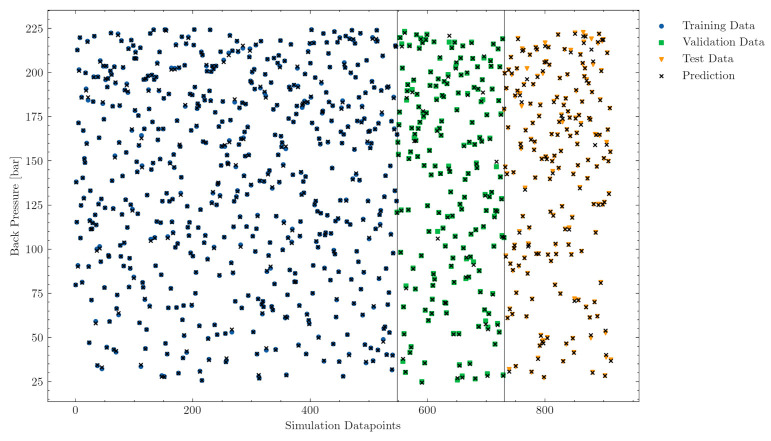
Accuracy of the back pressure predictions on the training (samples from which the model is trained), validation (unseen samples for hyperparameter tuning during training), and test (evaluation on unseen samples after training) data sets.

**Figure 9 polymers-13-02652-f009:**
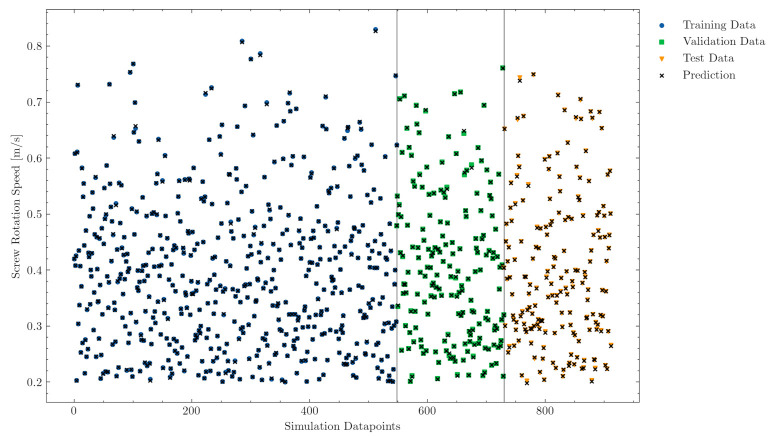
Accuracy of the screw rotational speed predictions on the training (samples from which the model is trained), validation (unseen samples for hyperparameter tuning during training), and test (evaluation on unseen samples after training) data sets.

**Figure 10 polymers-13-02652-f010:**
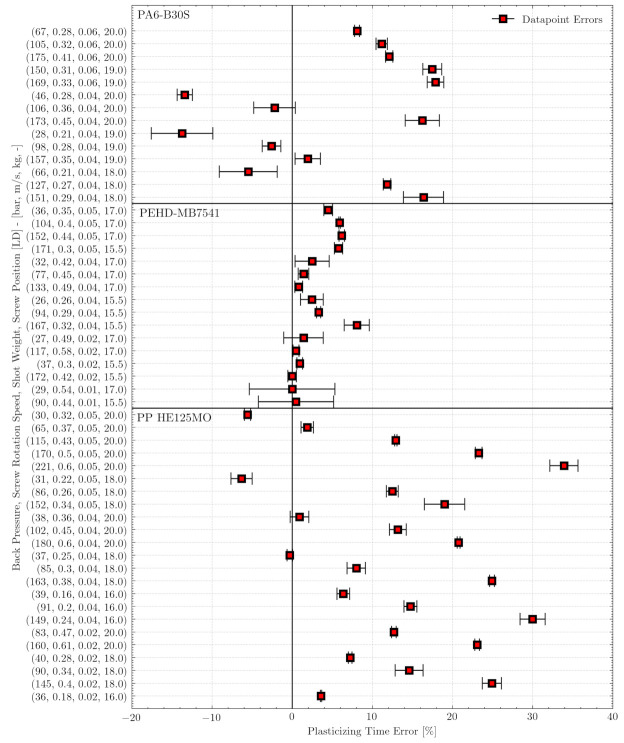
Mean error between the real and desired plasticizing times based on the predicted basic parameter settings obtained for three materials. Each scattered sample shows the mean and the standard deviation of three experiments per operating point. The mean absolute errors were 2.8%, 10.8%, and 14.5% for PEHD-MB7541, PA6-B30S, and PP-HE125MO, respectively. The ordinate shows the machine setting arrays for all experiments. An array contains the back pressures and screw rotational speeds predicted by the neural network model for specified shot weights and plasticizing times. The screw position where the material is 99% melted is described by the LD value. For example, the sample at the bottom (PP-HE125MO with the array (36, 0.18, 0.02, and 16)) shows a mean error of about 3% between the real and desired plasticizing times. The input information that the shot weight of 20 g is 99% melted at screw position LD 16 was fed into the neural network model, which predicted 36 bar back pressure and a 0.18 m/s screw rotational speed.

**Table 1 polymers-13-02652-t001:** Limits of the input parameters for the simulation. Within these limits, the data set was drawn randomly.

	Back Pressure	Metering Stroke	Screw Rotational Speed	Cycle Time
Min	25 bar	0.8 D	0.2 $\frac{m}{s}$	10 s
Max	225 bar	4 D	1 $\frac{m}{s}$	60 s

**Table 2 polymers-13-02652-t002:** Experimental configurations for 0.035 kg shot weight and 99% melt value. The first entry describes that, for a back pressure of 148.7 bar and a screw rotational speed of 0.24 m/s, 99% of 35 g of material is predicted to be melted at screw position LD 16 within a plasticizing time of 9.62 s.

Screw Position [LD]	Plasticizing Time [s]	Back Pressure [bar]	Screw Rotational Speed [$\frac{m}{s}$]
16	9.62	148.7	0.24
16	11.05	90.9	0.20
16	12.13	38.7	0.16
18	6.03	163.2	0.38
18	7.10	84.9	0.30
18	7.82	37.1	0.25
20	3.87	180.4	0.60
20	4.59	102.2	0.45
20	5.31	38.1	0.36

**Table 3 polymers-13-02652-t003:** Comparison of relevant supervised machine learning methods. The absolute prediction errors of the labels back pressure and screw rotational speed are listed for the training and test sets.

Method	Mean Error [%]		Std Error [%]	
	Train	Test	Train	Test
Multilayer Perceptron	0.21	0.27	0.26	0.37
Gaussian Process Regression	0.08	1.16	0.18	2.25
Polynomial Regression	0.34	1.27	0.55	4.98
Support Vector Regression	2.57	2.87	2.64	3.81
Random Forest	8.39	21.42	14.40	37.18
Gradient Boosting	18.44	24.34	31.44	43.54

**Table 4 polymers-13-02652-t004:** Performance of the neural network model.

Label	RMSE Train	RMSE Test
Back Pressure [bar]	0.41	0.61
Screw Rotational Speed [m/s]	0.0008	0.001

**Table 5 polymers-13-02652-t005:** Absolute errors between the real and desired plasticizing times for the predicted parameter settings. the mean and standard deviation were calculated based on all samples per material. Each maximum error was based on only one data point and gives further insights into the differences among the observations of each material.

	PP-HE125MO	PEHD-MB7541	PA6-B30S
Mean	14.4%	2.8%	10.8%
Std	10%	2%	6%
Max	34%	8%	18%

## Data Availability

The data presented in this study are available on request from the corresponding author.

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
