# Peer review of "A Simulation-Data-Based Machine Learning Model for Predicting Basic Parameter Settings of the Plasticizing Process in Injection Molding"

_polymers, 2021, doi:10.3390/polym13162652_

Round 1
Reviewer 1 Report
PAGE 2, lines 59-60: revised the sentence
page 5, lines 132-135: these 2 sentences do not clearly describe 2 input parameters, it seems more like one. However, further reading reveals these two parameters.
page 5, line 153: “therefore” requires comma. Correct throughout the text.
Page 5, line 156: Correct the figure number: instead of 2 it should be 4
page 7, line 202 add in brackets those two basic parameters
page 8: revised line 223 and the sentence “The.....is kept low" (lines 234-236.
page 8 line 239: the figure and the text are not consistent or the learning rate and loss are not the same parameters. Can you explain that? Also use either 1e-3 or 10-3 in both the text and figure.
page 8 line 241:add a reference.
figure 8 and 9 and the text related to the figures: define training, validation and test data.
The errors in table 5 are not consistent with those in figure 10. The mean errors for the samples PP HE125MO are lower than those of other two samples as shown in Figure 10. However, the numbers in figure 5 are showing the opposite. why?
Author Response
Here are my answers to your inputs and the diff-file with the changes. Note that there was something wrong with the new references. This issue is only visible in the diff-file, and not in the revised pdf-file.
PAGE 2, lines 59-60: revised the sentence
- Can you please be more precise what you mean?
page 5, lines 132-135: these 2 sentences do not clearly describe 2 input parameters, it seems more like one. However, further reading reveals these two parameters.
- done
page 5, line 153: “therefore” requires comma. Correct throughout the text.
- done
Page 5, line 156: Correct the figure number: instead of 2 it should be 4
- done
page 7, line 202 add in brackets those two basic parameters
- done
page 8: revised line 223 and the sentence “The.....is kept low" (lines 234-236.
- Can you please be more precise what you mean?
page 8 line 239: the figure and the text are not consistent or the learning rate and loss are not the same parameters. Can you explain that? Also use either 1e-3 or 10-3 in both the text and figure.
- Done; learning rate is different to loss. I think this should be clear from the text.
page 8 line 241:add a reference.
- done
figure 8 and 9 and the text related to the figures: define training, validation and test data.
- done
The errors in table 5 are not consistent with those in figure 10. The mean errors for the samples PP HE125MO are lower than those of other two samples as shown in Figure 10. However, the numbers in figure 5 are showing the opposite. why?
- There was an error with one horizontal line. I have fixed it and now there should not be any confusion. Thank you for telling me.
___________________________________________________________________________
I hope that everything (besides the two inputs I did not understood) is ok now for you.
A big thank you for reviewing the paper!
Best regards,
Schmid

Reviewer 2 Report
The authors have constructed a simulation-data-based multilayer perceptron model (or full-connect neural network) to predict (machine) settings for the plasticizing process in injection molding. They found that the predictions of the multilayer perceptron model outperformed other methods a lot. They also conducted experiments to evaluate their predictions on three different materials. I recommend this manuscript for publication in Polymers journal if authors can reply to the following comments properly.
1 There are several minor errors around eq. (1).First, summation in eq. (1) should be from j=1 to Q other than from j =0 . Second, From Line 88 to Line 93, variables "si", "ai", "xj" and "wij" should be "$s_i$", "$a_i$", "$x_j$" and "$w_{ij}$" (in latex math mode).
2 In Line 93, I suggest to delete "within the ranges [0,1] or [-1,1]", because there are activation functions (e.g. reLU function) that are NOT within the ranges [0,1] or [-1,1].
3 In Lines 99-100, the backpropagation algorithm is a very famous algorithm proposed (or popularized) by Rumelhart, Hinton and Williams in 1986, so it might be more propriate to cite their works other than ref. [8].
4 In the sample-genaration simulation, there are five random inputs (or 5D input vector). However, they generate only about training 549 samples to train the neural network. The author should convince us 549 samples are adequate to train a reliable model because there only 3.5 data points, on average, for each input dimension (3.5^5=549).
5 Authors should explain why the train loss is bigger than the validation loss in Figure 7.
Author Response
Here are my answers to your inputs and the diff-file with the changes. Note that there was something wrong with the new references. This issue is only visible in the diff-file, and not in the revised pdf-file.
1 There are several minor errors around eq. (1).First, summation in eq. (1) should be from j=1 to Q other than from j =0 . Second, From Line 88 to Line 93, variables "si", "ai", "xj" and "wij" should be "$s_i$", "$a_i$", "$x_j$" and "$w_{ij}$" (in latex math mode).
- done
2 In Line 93, I suggest to delete "within the ranges [0,1] or [-1,1]", because there are activation functions (e.g. reLU function) that are NOT within the ranges [0,1] or [-1,1].
- done
3 In Lines 99-100, the backpropagation algorithm is a very famous algorithm proposed (or popularized) by Rumelhart, Hinton and Williams in 1986, so it might be more propriate to cite their works other than ref. [8].
- Done (there is a problem the diff-version with the references. However, in the revised version it is correct
4 In the sample-genaration simulation, there are five random inputs (or 5D input vector). However, they generate only about training 549 samples to train the neural network. The author should convince us 549 samples are adequate to train a reliable model because there only 3.5 data points, on average, for each input dimension (3.5^5=549).
- Can you please be more precise what you mean by that? I have chosen 2000 samples (simulations) and have clearly shown by the performance of the model that the datapoints are more than enough.
5 Authors should explain why the train loss is bigger than the validation loss in Figure 7.
- done
___________________________________________________________________________
I hope that everything (besides the one input I did not understood) is ok now for you.
A big thank you for reviewing the paper!
Best regards,
Schmid

Reviewer 3 Report
In this work, the authors have clearly done good work by using simulation, machine learning methods and experiments to predict the settings for the plasticizing parameters back pressure and screw rotational speed. Some features of this manuscript are:
- Well-written in English
- Presenting of good and enough statistical data
- Presenting high quality images
However, this manuscript could be published in this journal after a few revisions as follows:
- Please describe the plasticizing process more. Does it mean the melting stage in polymer processes like extrusion and injection molding? Please move Figure 1 to the Introduction to clear plasticizing process for the readers.
- The introduction must be supported by more citations. Please cite the sentences below to the relevant articles:
- “As simulations become used increasingly for screw layout 26 and process optimization, this valuable information could be employed to determine basic machine settings.”
- “Many operators adjust the plasticizing parameters for process stability, but without additional knowledge about the current process.”
- “Due to the complex melting behavior of polymers, it is not exactly known whether a selected operating point is efficient.”
- A data-based digital twin of the plasticizing process that “knows” the correlations between melt quality and plasticizing parameters could therefore be very beneficial. Further, a simulation-data-based model could already be built in the screw-selection phase.
- “Digital twin” is still not clear to me. Could you please explain it more?
- It is true that the part quality depends on melt quality in the plasticizing process; however, melt behavior cannot control all part qualities. For instance, warpage is highly affected by mold temperature. So, please correct the following sentence:
- “However, problems that influence final part quality can already occur one step earlier, that is, in the plasticizing process, often due to insufficient melt quality.”
- Please provide a background of other commercial software and discuss their advantages and disadvantages in comparison with S3.
- It seems that S3 is an in-house software. Please introduce the software in a Supplementary File and provide some photos related to its interface as well as its numerical models.
- It is also recommended to compare S3 to the other commercial software like Ludovic® and XimeX-TSE:
- https://www.scconsultants.com/en/ludovic-twin-screw-simulation-software.html
- The comparison with other commercial software can be discussed in Introduction section
- Please also send a PDF file of ref [6] to me (and other reviewers), as I cannot access the document (thesis) through the University Library Linz.
- What does “melting behavior” mean as the output parameter in lines 67, 68 on page 2? How can you quantify and measure it? Please explain it on Page 2 rather than Page 5.
- Please support the statement below by citing to the literature:
- “Melt quality is measured by the percentage of molten material along the screw length. For example, the screw position (in length-to-diameter ratio; LD) at which 99% of the material is molten can be determined and used as input to the model in the form of two features.”
- In section “3.2. Model Construction”, all listed methods, architecture and hyperparameters should be represented in a table. Plus, it is recommended that the authors visualize the simple structure of each method in the Supplementary File.
- PP-HE125MO, PEHD-MB7541 and PA6-B30S are the trade names of the materials. The authors need to describe the materials too. Please provide more information about the materials’ molecular weight and melting temperature ranges.
- Following the above comment, I think the difference in average absolute errors (between real and desired plasticizing times) for PEHD-MB7541, PA6-B30S and PP-HE125MO is more related to materials’ rheological behavior. The higher average absolute errors may obtain for composites and polymeric blends. I suggest studying complex materials in future to improve the machine learning model for better prediction of plasticizing process settings
Author Response
Attached are my answers to your inputs. I hope that everything is clear for you now.
Many thanks for reviewing the paper!
Best regards,
Schmid

This manuscript is a resubmission of an earlier submission. The following is a list of the peer review reports and author responses from that submission.